# Molecular Mechanism of Small-Molecule Inhibitors in Blocking the PD-1/PD-L1 Pathway through PD-L1 Dimerization

**DOI:** 10.3390/ijms22094766

**Published:** 2021-04-30

**Authors:** Yan Guo, Yulong Jin, Bingfeng Wang, Boping Liu

**Affiliations:** Key Laboratory for Bio-Based Materials and Energy of Ministry of Education, College of Materials and Energy, South China Agricultural University, Guangzhou 510630, China; guoyan@stu.scau.edu.cn (Y.G.); wbfeng@scau.edu.cn (B.W.)

**Keywords:** PD-L1, BMS-200-related small-molecule inhibitor, molecular docking, molecular dynamics simulation, binding free energy

## Abstract

Programmed cell death-1 (PD-1), which is a molecule involved in the inhibitory signal in the immune system and is important due to blocking of the interactions between PD-1 and programmed cell death ligand-1 (PD-L1), has emerged as a promising immunotherapy for treating cancer. In this work, molecular dynamics simulations were performed on complex systems consisting of the PD-L1 dimer with (S)-BMS-200, (R)-BMS-200 and (MOD)-BMS-200 (i.e., S, R and MOD systems) to systematically evaluate the inhibitory mechanism of BMS-200-related small-molecule inhibitors in detail. Among them, (MOD)-BMS-200 was modified from the original (S)-BMS-200 by replacing the hydroxyl group with a carbonyl to remove its chirality. Binding free energy analysis indicates that BMS-200-related inhibitors can promote the dimerization of PD-L1. Meanwhile, no significant differences were observed between the S and MOD systems, though the R system exhibited a slightly higher energy. Residue energy decomposition, nonbonded interaction, and contact number analyses show that the inhibitors mainly bind with the C, F and G regions of the PD-L1 dimer, while nonpolar interactions of key residues Ile54, Tyr56, Met115, Ala121 and Tyr123 on both PD-L1 monomers are the dominant binding-related stability factors. Furthermore, compared with (S)-BMS-200, (R)-BMS-200 is more likely to form hydrogen bonds with charged residues. Finally, free energy landscape and protein–protein interaction analyses show that the key residues of the PD-L1 dimer undergo remarkable conformational changes induced by (S)-BMS-200, which boosts its intimate interactions. This systematic investigation provides a comprehensive molecular insight into the ligand recognition process, which will benefit the design of new small-molecule inhibitors targeting PD-L1 for use in anticancer therapy.

## 1. Introduction

PD-1 is a protein on the surface of activated T cells in humans, and the ligand PD-L1 expressed on many kinds of tumor cell. In the tumor microenvironment, high levels of PD-1 induce the expression of PD-L1 by causing the release of cytokines [1,2,3,4]. The upregulated PD-L1 is apt to bind with PD-1, which devitalizes T cells and impedes the immune response. Consequently, it allows cancer cells to grow wildly in the human body, leading to organ failure [5,6,7,8]. Based on this mechanism, blocking the binding of PD-1/PD-L1 could be a valid approach to reversing immunosuppressive conditions and freeing T cells to kill cancer cells effectively [9]. Recently, the first crystal structure of PD-1/PD-L1 was reported by Zak et al. [10], providing an accurate molecular basis for inhibitor design in this signaling pathway. Among the inhibitors, monoclonal antibodies (mAbs) have returned exciting results in clinical trials [11,12,13,14]. However, due to their high molecular weights, they possess several limitations, such as low steadiness and poor tumor-permeability profiles [13,15]. In comparison, small-molecule inhibitors (MW < 550 Da) can avoid these limitations and present a promising alternative way to disrupt the PD-1/PD-L1 axis [10,16].

In recent years, the BMS company claimed that two classes of active small-molecule inhibitors with different core scaffolds—2-methyl-3-biphenylyl and 2,3-dihydro-1,4-benzodioxinyl—can efficiently block PD-1/PD-L1 interactions, with IC_50_ values ranging from the nM to mM level [17]. Afterwards, Zak’s group disclosed the X-ray crystal complexes of several PD-L1 dimer/BMS inhibitors and deposited their coordinates and structural factors into the Protein Data Bank (PDB) [18,19,20]. Their results show that the inhibitors directly bind to PD-L1 and induce the formation of dimeric PD-L1, thus forming a hydrophobic tunnel pocket and interrupting the PD-1/PD-L1 signaling pathway [18,19,20]. In addition, structural details of the binding regions have been reported, showing that these compounds are located at CCʹFG strands of the PD-L1 dimer. Meanwhile, inside the dimer, BMS inhibitors interact with two distinct sites of PD-L1. These findings provide a basis for the further design of small-molecule compounds targeting PD-L1 [18,19,20,21]. However, the experimental studies above have not provided detailed descriptions of the ligand recognition process.

Combinations of molecular modeling tools, such as molecular docking, molecular dynamics (MD) simulation and the molecular mechanics–Poisson Bolzmann surface area (MM-PBSA) approach [22], offer us a unique opportunity to construct protein–small-molecule antagonist systems [23,24], analyze recognition mechanisms [25] and recognize key residues at binding pockets [25,26]. Recently, Huang et al. [27] predicted the hotspots in the PD-1/PD-L1 pathway and highlighted the importance of polar and nonpolar interactions between PD-1 and PD-L1 by MD simulations and binding free energy calculations. Ahmed et al. [28] evaluated the dynamic characteristics of small-molecule binding regions on PD-L1, emphasizing the specific flexible domains at the PD-L1 surface and the key role of methionine residue for binding. In terms of mAbs targeting this pathway, Sun et al. [29] and Shi et al. [30] claimed that the CC’FG area of PD-L1 is instrumental in the recognition of mAbs. In relation to small-molecule antagonists, molecular docking was applied by Almahmoud et al. [16] to evaluate hotspot residues on the PD-L1 dimer, showing that the residues Tyr56, Asp122 and Lys124 have important functions upon binding with BMS ligands. Similar results were also obtained by Shi et al. [31], who reported that the key residues on both PD-L1 monomers were Ile54, Tyr56, Met115, Ala121 and Tyr123. The studies above have devoted much effort to understanding the inhibition mechanism of BMS small-molecule targeting PD-L1. However, it is worth noting that although previous studies provide invaluable information, a full view of the molecular mechanism involved in the BMS ligand recognition process is still absent. Besides, the chirality of small-molecule inhibitors usually affects their inhibitory activities, and few works pay attention to this issue in relation to BMS molecules [32].

Accordingly, in this paper, on the basis of inhibitors related to BMS-200, we attempt to provide a molecular-based insight into how these small molecules block PD-1/PD-L1 interactions by combining a series of effective molecular modeling methods [24]. The original BMS-200, i.e., the S-enantiomer (PDB ID: 5N2F), was used as a modeling template and is denoted as (S)-BMS-200. Moreover, the R-enantiomer is named (R)-BMS-200, while (MOD)-BMS-200 is modified from the original (S)-BMS-200 (by replacing the hydroxyl group with a carbonyl to remove its chirality; see Figure 1). In relation to the PD-L1 dimer, we denote its monomers as _A_PD-L1 and _B_PD-L1, respectively, because of their unequal roles in the binding process. Molecular docking was first conducted to build PD-L1 dimer/(R)-BMS-200 and PD-L1 dimer/(MOD)-BMS-200 systems, which, hereafter, are referred to as R and MOD systems for convenience, while the PD-L1 dimer/(S)-BMS-200 is called the S system. Moreover, PD-L1 dimer and _A_PD-L1/(S)-BMS-200 systems were built as well. Then, MD simulation was carried out to investigate the detailed binding modes of these systems. Further, the binding free energy and per-residue energy decomposition calculations for these systems were compared and summarized systematically, with the aim of identifying energy contributions and key residues on the PD-L1 dimer. Finally, to understand how BMS-200-related molecules affect the overall dynamic characteristics of the PD-L1 dimer, we took advantage of free energy landscape (FEL) and protein–protein interaction (PPI) analyses of the S system, due to its relatively low binding free energy among such complex systems. Taken together, our work provides structural and energetic insights into the recognition process of BMS-200-related inhibitors, which will be useful in future explorations of efficient small-molecule inhibitors targeting the PD-1/PD-L1 pathway.

## 2. Results and Discussion

The BMS-200 small-molecule inhibitor (the original (S)-BMS-200) could block PD-1/PD-L1 interactions by inducing the dimerization of PD-L1, thereby activating the immune response of T cells and leading to tumor cell elimination by T cells [9]. To contrast the binding behaviors of BMS-200-related molecules (the original (S)-BMS-200, its R-enantiomer (R)-BMS-200, and the (MOD)-BMS-200 made without chirality by replacing the hydroxyl with the carbonyl from (S)-BMS-200) with the PD-L1 dimer, a series of molecular modeling approaches were used, including molecular docking, MD simulations and MM-PBSA calculations.

### 2.1. Docking

Docking software is considered reliable when it is able to generate a pose that is very close to the original pose in a crystal structure; i.e., the root mean square deviation (RMSD) between the docked pose and original pose is low (<1.5 or 2 Å) [33]. In this work, the overlap between the vina-docked pose and native conformation was as shown in Appendix A. The RMSD between the two poses was 0.77 Å. In addition to pose selection, interaction determination is another validation method. Here, a knowledge-based server, the Protein-Ligand Interaction Profiler (PLIP), was used to estimate the interaction mode of the docked pose. The results are in accordance with the study of Guzik et al. [20], which confirmed that Vina software can be used to define the binding conformations of other BMS-200-related ligands. The correlative results can be found in Appendix A.

### 2.2. RMSD

The convergence of the MD simulations was firstly determined in terms of the RMSDs of the receptors and ligands to qualitatively analyze their conformational stability at a scale of 150 ns. It can be seen that the RMSD values of the protein increased rapidly during the first 5 ns, then approached stability during the next 145 ns in the MOD, S and R systems (Figure 2a,b). Meanwhile, that of the protein in the dimer system fluctuated violently from 5 to 60 ns and became steady thereafter, indicating that the former had smaller conformational changes than the latter, which was expected due to the binding of inhibitors. Further, the PD-L1 dimer showed more flexible behaviors upon (R)-BMS-200 binding than (S)-BMS-200. The above results indicate that the MD products could be applied to further study. 

### 2.3. RMSF

To determine the effect of inhibitors on the overall PD-L1 target, all trajectory frames were used in root mean square fluctuation (RMSF) calculations. As shown in Figure 2c,d, β-sheet moieties showed high stability for both dimer and complex systems (around 2.0 Å), while loops showed higher RMSF values (up to 5.0 Å). The residues of the C, F and G regions fluctuated more in the dimer state (~2.0 Å) than in bound states (<2 Å), meaning that upon binding of BMS-200-related molecules, the domains showed more stable behavior. Our results agree with those of the literature, suggesting that sheet domains on the PD-L1 dimer are critical for the binding of small molecules and that the regions undergo significant conformational change [27,34]. In addition, when compared with the S system, more obvious fluctuations could be observed in the C”D loop of _A_PD-L1 and the BC loop of _B_PD-L1 in the R system, implying that the latter has enhanced flexibility, which was also coincident with the RMSD values.

### 2.4. Binding Free Energy

To further analyze the binding affinities of the PD-L1 dimer and BMS-200-related molecules, the binding free energies (Δ*G*) were evaluated via the g_mmpbsa program.

From Table 1, it can be seen that the Δ*G* values of the MOD, S and R systems were −42.45 ± 0.35, −42.42 ± 0.21 and −40.48 ± 0.21 kcal/mol, respectively, indicating that the PD-L1 dimer had similar affinities for (S)-BMS-200 and (MOD)-BMS-200, but a slightly weaker binding affinity for (R)-BMS-200. Compared to the complex systems, the dimer system presented a positive Δ*G* (36.11 ± 0.89 kcal/mol), while the ∆*G* (−20.17 ± 2.20 kcal/mol) of the _A_PD-L1/(S)-BMS-200 system was roughly less than half of that of the S system, suggesting that PD-L1 can hardly be spontaneously dimerized at all, and that BMS-200-related molecules are indispensable for the dimerization of PD-L1.

According to the MM-PBSA method, ΔG can be divided into polar (Δ*E*_polar,total_) and nonpolar (Δ*E*_nonpolar,total_) energies. Then, the contributions of the nonpolar and polar interactions can be further decomposed into *E*_vdw_ + *E*_SA_ and *E*_ele_ + *E*_PB_, respectively. As shown in Table 1, the total nonpolar interactions were the major driving force for the binding process in all systems, while the total polar contribution was unfavorable, mainly due to the effect of the solvent (*E*_PB_). This finding agrees with the results reported by Shi et al. [31], in which BMS-8 and BMS-1166 mainly formed nonpolar interactions with side chains of the residues on PD-L1 monomers.

### 2.5. Per-Residue Energy Decomposition

The residues with important roles in the binding of complex systems were identified via per-residue energy decomposition. As shown in Figure 3, the key residues in the MOD system included _A_Ile54, _A_Tyr56, _A_Met115, _A_Tyr123, _B_Ile54, _B_Tyr56, _B_Val68, _B_Met115, _B_Ala121 and _B_Tyr123; the key residues in the S system included _A_Ile54, _A_Tyr56, _A_Met115, _B_Ile54, _B_Tyr56, _B_Val68, _B_Met115, _B_Ala121, and _B_Tyr123; the key residues in the R system included _A_Ile54, _A_Tyr56, _A_Met115, _A_Tyr123, _B_Ile54, _B_Tyr56, _B_Val68, _B_Met115, _B_Ala121, and _B_Tyr123. In short, the key residues Ile54, Tyr56, Met115, Ala121, and Tyr123 on both monomers participated in the recognition of BMS-200-related molecules. Our results are in qualitative agreement with those in the literature, where the key residues were found to be Ile54, Tyr56, Met115, Ala121, and Tyr123 on both PD-L1 monomers in PD-L1/small-molecule systems [21,31,34]. In addition, the energy decomposition results also show that Met115 exhibited obvious and constant contributions to binding in all of the complex systems. This is consistent with the results of Ahmed et al. [28] and Lim et al. [35], who argued that the presence of small-molecule inhibitors facilitates the conformational change of Met115, resulting in its improved accessibility to binding sites. Moreover, it is worth noting that the key residues of _A_PD-L1 only contributed about 12% of the binding free energy to complex systems, while more (ca. 23%) was devoted by _B_PD-L1. This finding agrees with the published data, showing that both BMS-8 and BMS-1166 inhibitors tend to have a more stable binding mode with one monomer than the other [36]. Based on the equal Δ*G* values of the MOD and S systems, calculations were only carried out for the S and R systems in the following section.

### 2.6. Contact Numbers 

To further identify whether (S)-BMS-200 and (R)-BMS-200 preferentially interacted with certain residues of the PD-L1 dimer, the average interchain contact numbers were calculated. Herein, the same criterion of 10 contacts used in the literature was employed to identify the residues with significant effects on the interactions [36]. From Figure 4, six binding domains were identified, including the N-terminal (residues 18–20), C sheet (residues 54–56), C” sheet (residues 66–68), loop (residues 75–76), F sheet (residues 115–117) and G sheet (residues 121–124).

The corresponding contact numbers of these domains are listed in Table 2. Overall, the total contact number was slightly higher in the S system, which is also in accordance with the calculated binding free energy. Specifically, it was clear that sheet domains contributed the majority of contact numbers for both S and R systems, though that for the latter was slightly lower. As shown in Figure 5, the most striking features for this region came from the intimate hydrophobic interactions between the sheet domains and aromatic rings of both (S)-BMS-200 and (R)-BMS-200. In addition, in the sheet domain of the R system, H bonds could be detected between the charged residues, including _A_Asp122 and _A_Lys124, and the hydrophilic tail of (R)-BMS-200, such as protonated nitrogen and hydroxybutyric acid. However, such interactions were absent in the S system. For the N-terminal domain, moderate interactions with contact numbers of 33 and 22 could be observed for the S and R systems, respectively. It was found that the most contributions came from the binding between _A_Ala18 or _A_Phe19 with the inhibitors in both systems. The H bond between the O5 in (S)-BMS-200 and the side chain of _A_Phe19 further stabilized the interchain interactions of the S system. The interactions originating from the loop domain were comparable or even weaker than those in the N-terminal domain. It was found that the inhibitors would interact with this domain via _B_Lys75 and _B_Val76, and the contact between the positively charged residue _B_Lys75 and (R)-BMS-200 was stronger than that with (S)-BMS-200. These results are consistent with the binding free energy results—that electrostatic interactions (*E*_ele_) contribute more in the R system than in the S system (see Table 1).

### 2.7. Nonbonded Interactions

Based on the binding modes and structures of the inhibitors (Figure 1), we can infer that H bonds and hydrophobic interactions provide much of the binding in BMS-200-related inhibitors and the PD-L1 dimer. Similarly, Shi et al. [31] and Sun et al. [37] argued that H bonds and hydrophobic interactions are crucial to BMS inhibitors used in anticancer treatment. As a result, we first made a quantitative evaluation of the occupancy of intermolecular H bonds. From Table 3, it can be seen that the O3 in the side chain of (S)-BMS-200 prefers to link with _B_PD-L1 by H bonds and not with _A_PD-L1, while the opposite trend is observed in the R system (Table 4). Meanwhile, _A_Phe19 forms a H bond (occupancy = 28.57%) with the O5 of (S)-BMS-200, while a H bond with an occupancy of 42.52% could be seen between _A_Asp122 and the protonated nitrogen of (R)-BMS-200. The results are consistent with those of previous studies, in which the importance of _A_Phe19 and _A_Asp122 in the stabilization of complexes is highlighted [30,31]. Meanwhile, it indicates that H bonds between charged residues and (R)-BMS-200 are more likely to come into being. However, _A_Asp122 was not identified as a key residue in the R system, implying that H bonds probably do not play a dominant role in the binding process, though several polar residues like Gln66, Asp122 and Lys124 could be found in the pocket, which agrees with the results obtained by Sun et al. [29].

Next, the hydrophobic interactions were also evaluated. As shown in Figure 5, such interactions were obvious in both systems, in which (S)-BMS-200 and (R)-BMS-200 were bound in a long cylindrical pocket inserted deep into the interface between _A_PD-L1 and _B_PD-L1, while the dihydro-benzodioxinyl group of inhibitors was located at the very bottom of the pockets. The pocket of the S system was surrounded by the side chains of _A_Phe19, _A_Ile54, _A_Tyr56, _A_Met115, _A_Ala121, _A_Tyr123, _B_Ile54, _B_Tyr56, _B_Ala66, _B_Val68, _B_Met115, _B_Ala121, and _B_Tyr123. The side chain of _A_Phe19 extended towards the (S)-hydroxybutyric acid moiety; other residues approached the aromatic rings of (S)-BMS-200. The binding pocket of the R system was found to be surrounded by _A_Ile54, _A_Tyr56, _A_Met115, _A_Ala121, _A_Asp122, _A_Tyr123, _A_Lys124, _B_Ile54, _B_Tyr56, _B_Val68, _B_Met115, _B_Ala121, and _B_Tyr123. In detail, side chains of _A_Asp122 and _A_Lys124 were close to the (R)-hydroxybutyric acid moiety of (R)-BMS-200, and the side chains of other residues were adjacent to its aromatic rings. The interacting modes given in this work are consistent with results reported for BMS small-molecule and [1,2,4] triazolo [4,3-a] pyridines inhibitors [31,34,37]. Importantly, the residues Ile54, Tyr56, Met115, Ala121, and Tyr123 were highly conserved and located around the binding sites. Together with the previous energy decomposition result—that the same residues also contribute much to binding with BMS-200-related inhibitors (Figure 3)—it implies that these residues are promising targets for the development of drugs that can efficiently block PD-1/PD-L1 interactions. On the other hand, it was also found that the binding process was highly dependent on hydrophobic interactions between the aromatic rings of BMS-200-related inhibitors and hydrophobic residues such as _A_Ile54, _A_Met115, _A_Ala121, _B_Ile54, _B_Val76, _B_Met115, and _B_Ala121. Likewise, via the QSAR method, Shi et al. [31] revealed that the substituents at the aromatic rings of BMS inhibitors have great effects on their inhibitory activities. Thus, it implies that, with respect to BMS small-molecules, structural modification of the aromatic rings could possibly be a preferable way to improve their efficiency in inhibiting the PD-1/PD-L1 pathway.

In summary, according to the binding free energy results, there were no significant differences in the energy values of the MOD and S systems during the timescale of the simulations, while the R system showed a slightly higher value. Moreover, according to the nonbonded interactions, (R)-BMS-200 was more likely to form H bonds with charged residues than (S)-BMS-200. On the other hand, these complex systems exhibited similar binding modes; among them, hydrophobic interactions were the dominant stabilizing factor. Consequently, in the following section, only the results of the S and dimer systems are discussed to further elucidate the effect of (S)-BMS-200 on the global dynamics of the PD-L1 dimer.

### 2.8. Effects of (S)-BMS-200 on the PD-L1 Dimer

In order to monitor the mobility of the S and dimer systems more accurately, cross-correlation analysis was carried out to evaluate the detailed atomic dynamic state of the PD-L1 dimer [38]. In Figure 6, highly positive regions (red) indicate strong correlations with residue motion, while negative regions (blue) indicate anticorrelations. The depth of color on the diagonal represents the degree of motion of the atoms [24].

Figure 6 shows stronger motion of the residues in the two systems on the diagonals, especially those on the C-terminal of both PD-L1 monomers, which is attributed to the high flexibility of the residues on these domains. The fully filled blue patches in Figure 6a clearly show that anticorrelated motions of the amino acid residues between _A_PD-L1 and _B_PD-L1 were obvious in the dimer system, while such a phenomenon could hardly be observed in the S system. It suggests that the S system exhibited more stable dynamic behavior than the dimer system, owing to the linkage of (S)-BMS-200.

Then, the two largest principal components (PC1 and PC2) were utilized as reaction coordinates to graph the FEL. The blue regions represent a higher probability of conformational distribution associated with lower free energy [29]. As shown in Figure 7, the S system had only two independent large-area basins with relatively low free energy, while a greater number of smaller basins could be found in the dimer system. In addition, the shapes of the two FELs were significantly different, and the conformational transition space of the dimer system was larger than that of the S system. This indicates that the conformation of the PD-L1 dimer became less flexible due to stabilization by (S)-BMS-200.

Next, we studied the effect of (S)-BMS-200 on the structure of the PD-L1 dimer. As shown in Figure 8, compared with the PD-L1 dimer system, rearrangements of the dynamic structures, especially regions where (S)-BMS-200 was bound, could be observed throughout the whole simulation process in the S system. In addition, though it had subtle conformational changes, the PD-L1 dimer remained stable in the absence of (S)-BMS-200. This indicates that rearrangement of the dimer in the S system was induced by (S)-BMS-200 and that the stability of the PD-L1 dimer was key to the inhibitory mechanism of the (S)-BMS-200 ligand. Furthermore, as shown in Figure 9, the distances between the residue pairs of _A_Ile54/_B_Ile54, _A_Tyr56/_B_Tyr56 and _A_Met115/_B_Met115 in the S system were smaller than those in the dimer system. However, larger distances between residue pairs of _A_Ala121/_B_Ala121 and _A_Tyr123/_B_Tyr123 could be observed in the S system. This reveals that the binding domains in one PD-L1 monomer moved to the nearby regions of the other PD-L1 monomer, thus forming a more stable binding pocket in the presence of (S)-BMS-200.

To further investigate the conformational changes in residues in binding regions, a representative structure of MD in the dimer system was generated and superimposed on the PD-L1 dimer in the S system. As shown in Figure 10, in the presence of (S)-BMS-200, it could be found that the residues _A_E58, _A_V111, _A_A121, _A_Y123, _A_R125, _B_Y56, _B_D61, _B_V111, _B_R113, _B_M115, _B_G120, _B_A121, _B_Y123, and _B_R125 exhibited evident conformational changes with dramatically varying RMSF values (Figure 2c,d), particularly for _A_V111, _A_A121, _A_Y123, _B_V111, _B_R113, _B_M115, _B_G120, _B_A121, and _B_Y123. More importantly, some of them were key residues in the S system.

Then, the simplified PPI networks between _A_PD-L1 and _B_PD-L1 were built based on a series of nodes and edges representing residues and interactions, respectively. As shown in Figure 11, the PPI residues between the two PD-L1 monomers in the dimer system were composed of _A_E58, _A_D61, _A_V111, _A_R113, _A_A121, _A_Y123, _A_R125, _B_Y56, _B_D61, _B_V111, _B_R113, _B_M115, _B_G120, _B_A121, _B_Y123, and _B_R125; whereas in the S system, the residues _A_C61, _A_V111, _A_Y123, _A_R125, _B_Y56, _B_E58, _B_C61, and _B_R125 were involved in the PPI. A comparison of the PPI residues of the two systems showed that the residues _A_E58, _A_V111, _A_A121, _A_Y123, _A_R125, _B_Y56, _B_D61, _B_V111, _B_R113, _B_M115, _B_G120, _B_A121, _B_Y123, and _B_R125 were specific to the dimer system only, which suggests that the PPI sites of the PD-L1 dimer rearranged upon (S)-BMS-200 binding.

In addition to the residues mentioned above, the side chain of _A_Y56 has also been reported to exhibit significant rearrangement after binding with BMS ligands [10,20,39]. Hence, the motion direction of _A_Y56 in both systems was determined by calculating the dihedral angles φ and ψ, the distance between the CZ and O atoms, and the angle of benzene-CB-CA [37]. The dihedral angle variations (φ and ψ) of _A_Y56 in both systems are shown in Figure 12. The radii of the circular charts represent the simulation time (total = 150 ns). They show that the dihedral angle (φ) of _A_Y56 fluctuated from 189° to 275° in the dimer but ranged from 183° to 259° in the S system. As to the dihedral angle (ψ) of _A_Y56, it fluctuated from 80° to 167° in the dimer but from 96° to 173° in the S system. The narrower range of dihedral angle fluctuation in the S system demonstrates that _A_Y56 played a vital role in molecular recognition and flexibility reduction in the C-sheet domain. In addition, the benzene-CB-CA angle remained almost the same from 110° to 125° in both systems. The distance (CZ-O) was maintained at 5 Å in the dimer system but increased to 6 Å in the S system. Taken together, it was speculated that when PD-L1 or the PD-L1 dimer bind with (S)-BMS-200, the benzene ring of _A_Y56 shifts slowly towards the dihydro-benzodioxinyl group of (S)-BMS-200 around the CA-CB axis, rather than sharply flipping to the final position. This may allow this fragment of _A_Y56 to be more accessible to solvent, thus leading to the formation of a hydrophobic tunnel.

In summary, BMS-200-related inhibitors partially covered the PD-1 binding regions on each PD-L1 monomer. Furthermore, the interactions between two PD-L1 molecules also occupied the binding interface between PD-1 and PD-L1, thereby preventing PD-1/PD-L1 interactions. Based on the inhibitory principle proposed by Sun et al. [37], a possible inhibition mechanism of BMS-200-related compounds is proposed below. First, BMS-200 molecules recognize PD-L1 to form a PD-L1_BMS-200 complex, then they recruit another PD-L1 monomer to form a PD-L1 dimer_BMS-200 complex with improved stability. The plasticity of the PD-L1 surface could provide further potential for the rational design of small-molecule drugs targeting the PD-1/PD-L1 pathway.

## 3. Computational Details

### 3.1. Molecular Models

The three-dimensional structure of (S)-BMS-200 originated from PDB (ID: 5N2F, S-enantiomer). For (R)-BMS-200 and (MOD)-BMS-200, due to the lack of crystal parameters, their initial structures were first drawn in Gauss View 5.0 software [40] and then optimized to the minimized energy in Chem 3D software using the MM2 forcefield (see Figure 1) [16]. The missing parts of the PD-L1 structures were completed using the WHAT IF server.

### 3.2. Molecular Docking

AutoDock Vina is a new program for molecular docking that aims to predict the bound conformations and binding affinity of a complex [41]. In this work, it was used for automatic placement of (R)-BMS-200 and (MOD)-BMS-200 in the binding pocket of the PD-L1 dimer to obtain the initial structures of the R and MOD systems (Figure 13). For molecular docking, a grid of 20 × 20 × 20 points was established along the *x-*, *y-* and *z*-axes with a grid spacing of 1 Å. Its center was set as the binding pocket center; the other parameters were kept at the default settings. Among the poses obtained from docking, only the complexes with the lowest energy were selected for further MD simulation and analysis.

### 3.3. Molecular Dynamics Simulation

After preliminary docking results were obtained for the BMS-200-related molecules, the General Amber Force Field (GAFF) [42] parameters were created utilizing the Antechamber program in the AmberTools package [43]. Pertaining to PD-L1, the Amber ff99SB force field was adopted [44]. Each complex was placed into a cubic box with a distance of >10 Å to the solute; then, TIP3P waters were added and replaced by an appropriate number of counterions to maintain electroneutrality under physiological conditions.

All MD simulations were carried out using the GROMACS 2016.4 package [45]. Initially, to remove bad interactions and atomic collisions, energy minimization was performed, including steepest descent and conjugated gradient calculations. Later, the systems were heated gradually from 0 to 300 K over a timescale of 1 ns in the NVT ensemble and then equilibrated under 1 atm pressure for 1 ns in the NPT ensemble, where the V-rescale scheme and Parrinello–Rahman barostat were chosen to control the temperature and pressure, respectively. Finally, MD simulations were performed for 150 ns, during which the length of the hydrogen bond (H bond) was constrained using the LINCS algorithm. All short-range nonbonded interactions were cut off at 10 Å and long-range electrostatic interactions were calculated with the Particle Mesh Ewald (PME) method. Newton’s classical equations of motion were integrated using the Verlet Leapfrog algorithm and the trajectories were collected at time steps of 1.0 ps for subsequent analysis.

### 3.4. Binding Free Energy Calculation

Following the MD simulations, the binding free energies of the systems were calculated using the MM-PBSA approach [46] implemented in the GROMACS 2016.4 package.

Given the high computational costs, a total of 300 stable trajectory snapshots were extracted from 120 to 150 ns with a time interval of 100 ps. For all conformations, the binding free energy (Δ*G*) was obtained as below: (1)$${\Delta G=G}_{\mathrm{complex}\;}{-\;(G}_{\mathrm{PD}-L1\;}{+G}_{\mathrm{BMS}-200})$$
(2)$${\Delta G=\Delta E}_{\mathrm{gas}}{+\Delta G}_{\mathrm{sol}}\;–\;T\Delta S$$
(3)$$E_{\mathrm{gas}}{=E}_{\mathrm{vdw}}{+E}_{\mathrm{ele}}$$
(4)$$G_{\mathrm{sol}}{=E}_{\mathrm{PB}}{+E}_{\mathrm{SA}}$$
(5)$$E_{\mathrm{SA}}=\gamma \cdot \mathrm{SASA}$$
where *G*_complex_, *G*_PD-L1_ and *G*_BMS-200_ are the free energies (*G*) of PD-L1 dimer/BMS-200, PD-L1 dimer and BMS-200, respectively, being the average values of the extracted snapshots. G could be decomposed into the gas-phase energy (*E*_gas_), solvation-free energy (*G*_sol_) and the entropy contribution (*T*Δ*S*). *E*_gas_ consisted of a van der Waals term (*E*_vdw_) and electrostatic term (*E*_ele_). *G*_sol_ could be further divided into polar solvation energy (*E*_PB_) and nonpolar solvation energy (*E*_SA_). Here, the effect of entropy change (Δ*S*) was neglected for the following reasons. First, the purpose of this work was to compare the relative binding free energies of these systems but not their absolute values. In addition, many studies suggest that the MM-PBSA method can attain high computational accuracy, even without considering entropy change [47,48,49,50]. Moreover, the computed entropy was impacted by fewer conformations, and the entropic calculations were time consuming and computationally expensive [50].

To obtain the energy contributions of the key residues on both PD-L1 monomers, a per-residue energy decomposition analysis was also performed via the MM-PBSA method.

### 3.5. Simulation Analysis

The MD products were estimated using additional programs provided in the GROMACS 2016.4 package. The contact number between the PD-L1 dimer and BMS-200-related inhibitors was estimated by the mindist program with a cut-off of 6.0 Å. Furthermore, the occupancies of interchain H bonds in the S and R systems were counted using Visual Molecular Dynamics (VMD) 1.9.3. software [51] with the common standard; i.e., an acceptor–hydrogen-donor angle > 135° and an acceptor–hydrogen atom distance of <3.5 Å.

Principal component analysis (PCA) is a valid tool for calculating the major motion of protein atoms via projection of their trajectories along the directions described by selected eigenvectors [52]. It was used to describe the relativity of atomic motion throughout the MD simulation process. However, a standard PCA cannot distinguish internal motion from trivial overall motion very well [53,54]. To avoid this problem, PCA was carried out using the backbone dihedral angles of the PD-L1 dimer (dihedral PCA). The largest two principal component eigenvectors (dihedral PC1 and PC2) were used as the reaction coordinates to build the FEL [23,55]. Using this procedure, the effects of (S)-BMS-200 on the overall dynamics of the PD-L1 dimer could be directly revealed in terms of energy.

To further determine the effect of (S)-BMS-200 on the structure of the PD-L1 dimer, the PPI between _A_PD-L1 and _B_PD-L1 was investigated in the S and dimer systems. To this end, a simpler network model was built with the online server RING [56] using the MD products of these systems. The topology and node file represented residue and interaction types, respectively, and interactions such as H bonding, van der Waals forces and ionic bonding were taken into account. Then, the PPI networks were visualized using Cytoscape [57]. Finally, the vital PPI residues between two PD-L1 monomers were further extracted from the networks using the RIN analyzer within Cytoscape software.

## 4. Conclusions

In this work, based on the high-resolution (1.7 Å) crystal structure, together with multiple molecular modeling methods, we provided a detailed and comprehensive insight into the inhibitory mechanism of BMS-200-related small-molecule inhibitors. These include the original (S)-BMS-200, its R-enantiomer (R)-BMS-200, and the nonchiral (MOD)-BMS-200 formed by replacing the hydroxyl with a carbonyl at the chiral center of (S)-BMS-200. According to the binding free energy and per-residue energy decomposition results, formation of the PD-L1 dimer was promoted by BMS-200-related inhibitors. The MOD system exhibited the same binding free energy as the S system, while the R system had a slightly higher value. The key residues on both PD-L1 monomers of these systems are Ile54, Tyr56, Met115, Ala121, and Tyr123. Through the analysis of contact numbers and interactions, three main binding domains were identified, including the C, F and G strands of the PD-L1 dimer, and the dominant role of nonpolar interactions in stabilizing the S and R systems was highlighted. In addition, compared with (S)-BMS-200, (R)-BMS-200 tended to form H bonds with charged residues. Furthermore, FEL and PPI analyses revealed that (S)-BMS-200 can bind with the PD-L1 dimer stably and induce significant conformational changes in the key residues on the PD-L1 dimer, thus accelerating compact interactions. In conclusion, the results of this study will be useful in the structural modification and design of small-molecule inhibitors targeting the PD-1/PD-L1 pathway.

## Figures and Tables

**Figure 1 ijms-22-04766-f001:**
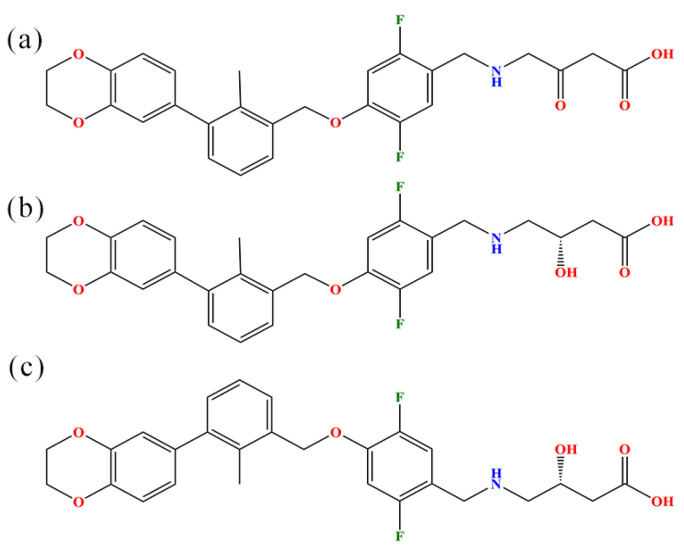
Chemical structural formulas of (**a**) (MOD)-BMS-200, (**b**) (S)-BMS-200 and (**c**) (R)-BMS-200.

**Figure 2 ijms-22-04766-f002:**
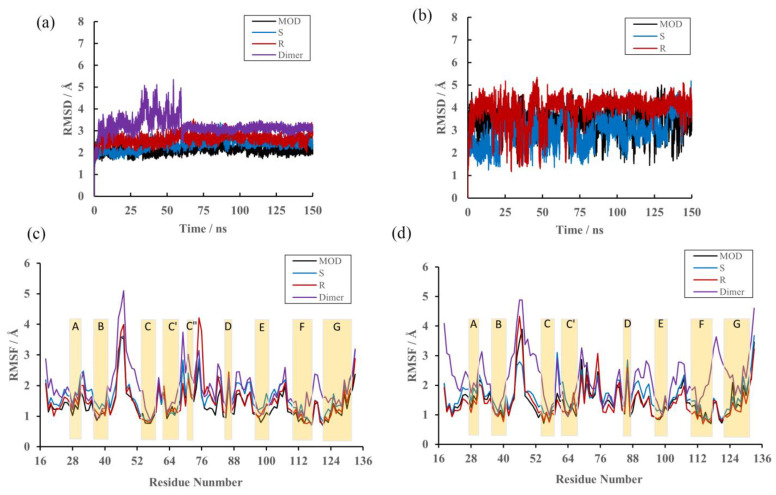
RMSD and RMSF results of the MD simulations. (**a**) RMSDs of backbone atoms in the MOD, S, R and dimer systems. (**b**) RMSDs of heavy atoms in (MOD)-BMS-200, (S)-BMS-200 and (R)-BMS-200. (**c**) RMSF fluctuations of residues on _A_PD-L1. (**d**) RMSF fluctuations of residues on _B_PD-L1.

**Figure 3 ijms-22-04766-f003:**
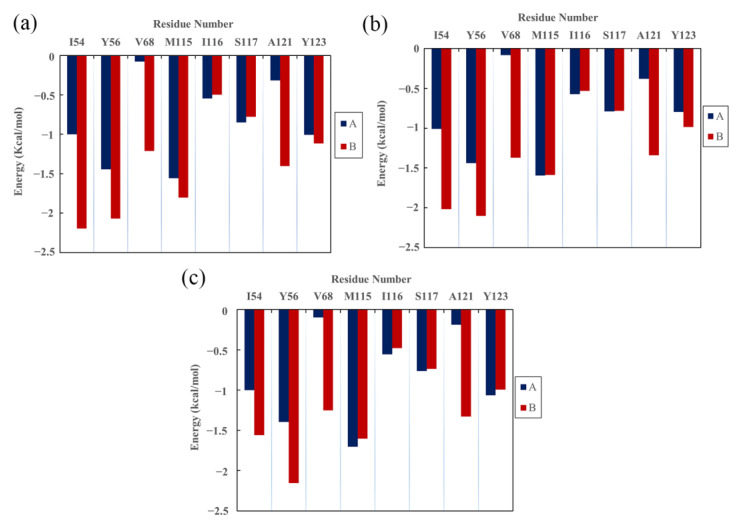
Residue energy decomposition of key residues belonging to the (**a**) MOD, (**b**) S and (**c**) R systems.

**Figure 4 ijms-22-04766-f004:**
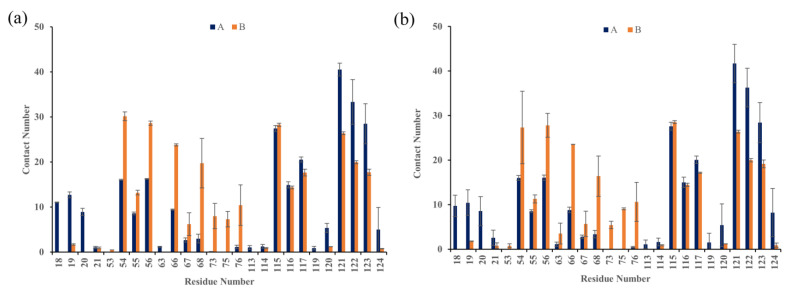
Contact numbers between BMS-200-related inhibitors and the PD-L1 dimer in the (**a**) S and (**b**) R systems. The error bars represent the standard deviations of 3 repeated calculations.

**Figure 5 ijms-22-04766-f005:**
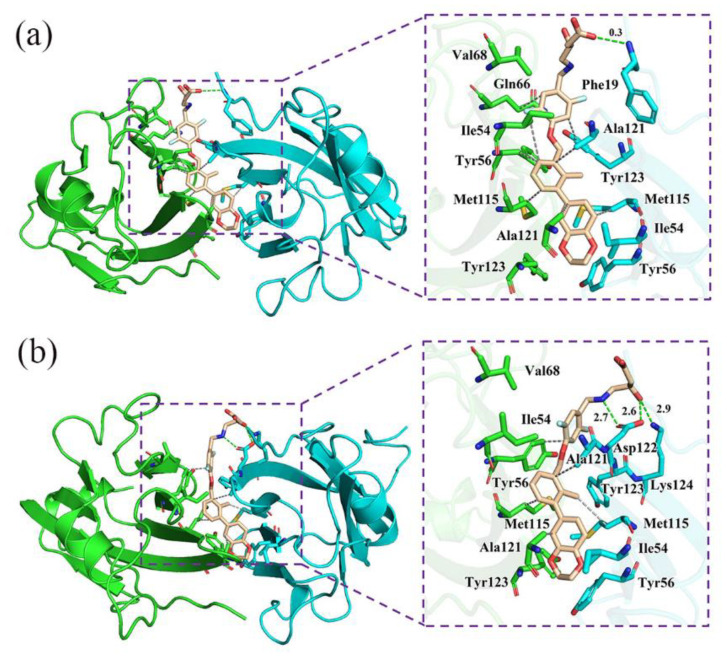
The binding modes of the (**a**) S and (**b**) R systems. The key residues on _A_PD-L1 and _B_PD-L1 are shown as cyan and green sticks, respectively, while the ligands are shown as beige sticks. The hydrophobic interactions and H bonds are shown as grey and green dashes, respectively.

**Figure 6 ijms-22-04766-f006:**
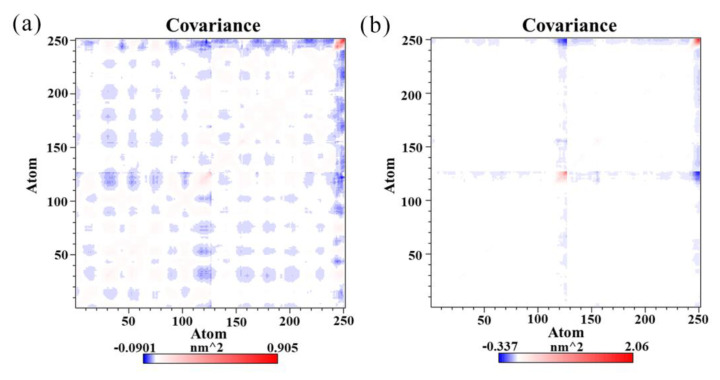
Cross-correlation matrixes of fluctuations in the *x*-, *y*-, and *z*-coordinates for Cα atoms belonging to the PD-L1 dimer in the (**a**) dimer system and (**b**) S system.

**Figure 7 ijms-22-04766-f007:**
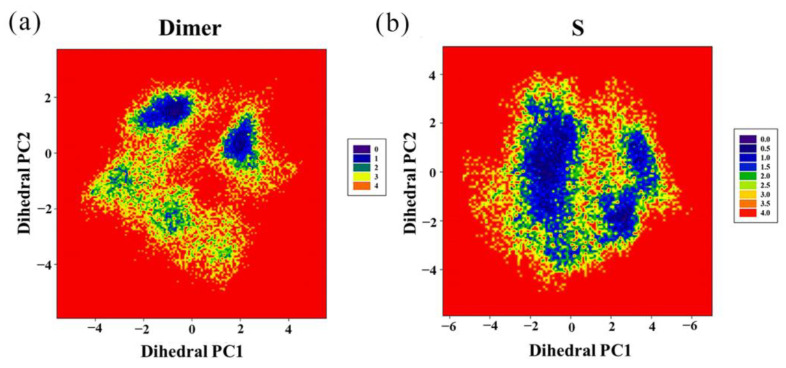
Free energy landscapes (KT) of the (**a**) dimer system and (**b**) S system. PC1 and PC2 represent principal components 1 and 2, respectively.

**Figure 8 ijms-22-04766-f008:**
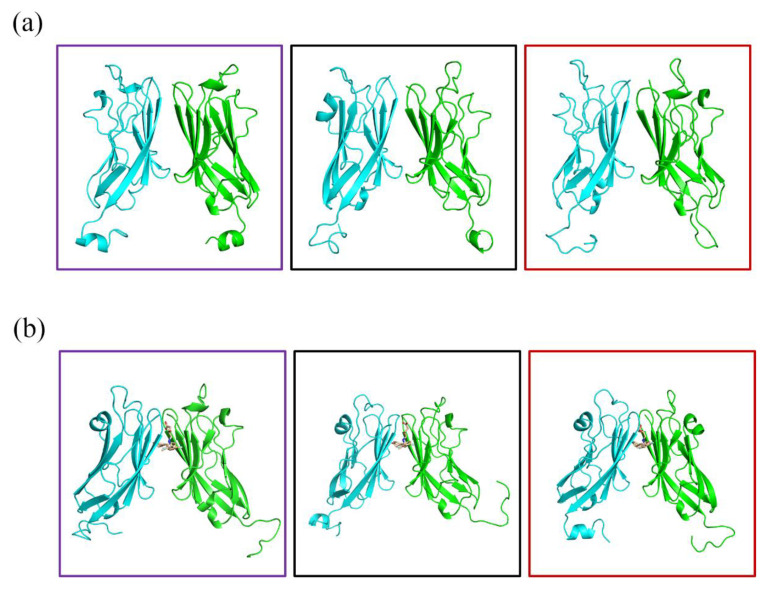
Structural dynamics of the (**a**) dimer and (**b**) S systems over the 150 ns MD simulation period. Violet boxes show the starting structures averaged at 0–50 ns, black boxes show averaged intermediate structures at 50–100 ns, while red boxes show the averaged end structures at 100–150 ns.

**Figure 9 ijms-22-04766-f009:**
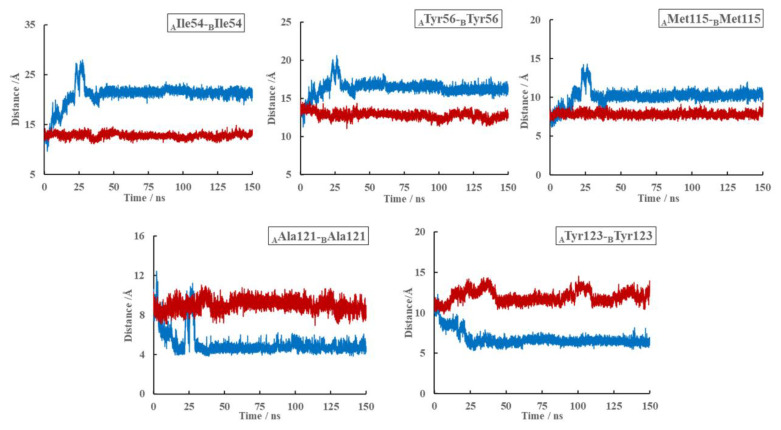
Distances of the key residue pairs of the dimer (blue) and S (red) systems over the whole 0–150 ns MD simulation period.

**Figure 10 ijms-22-04766-f010:**
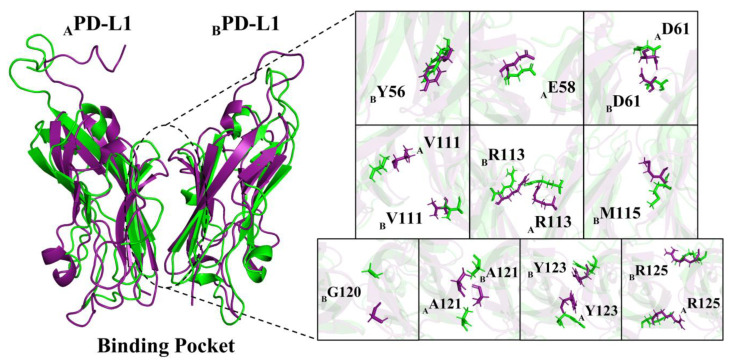
Residues with evident conformational changes in the dimer (purple) and S (green) systems.

**Figure 11 ijms-22-04766-f011:**
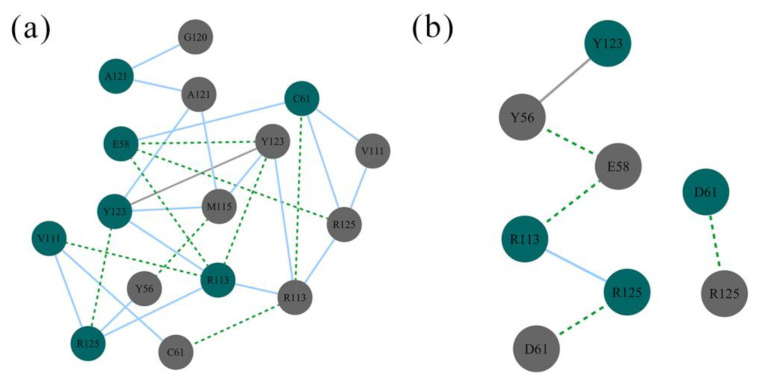
PPI networks of the (**a**) dimer and (**b**) S systems. Residues on _A_PD-L1 and _B_PD-L1 are represented as green and grey nodes, respectively. Hydrogen bonding, van der Waals, and π-π stacking interactions are depicted by green dashed, blue solid, and grey solid lines, respectively.

**Figure 12 ijms-22-04766-f012:**
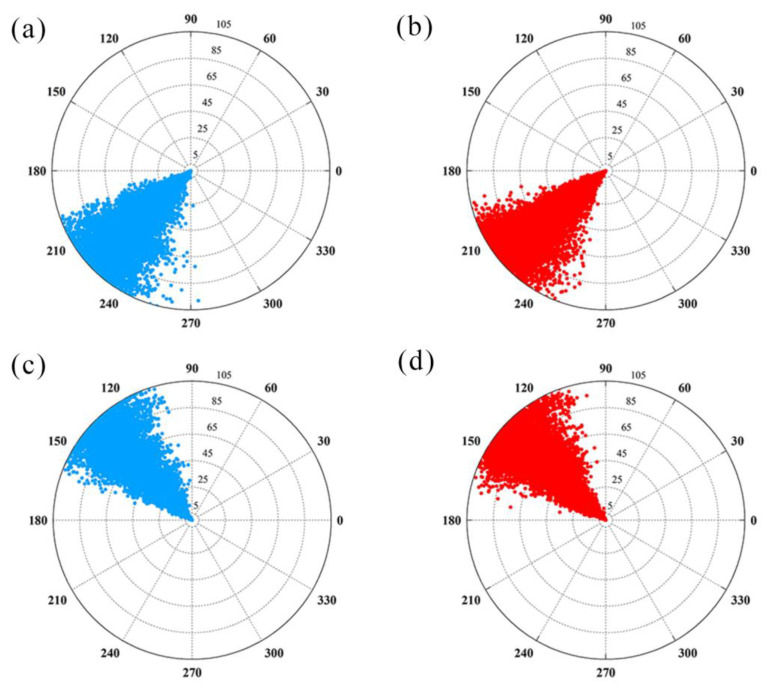
Structural comparison of _A_Y56 in the dimer and S systems. (**a**) φ in the dimer system. (**b**) φ in the S system. (**c**) ψ in the dimer system. (**d**) ψ in the S system. (**e**) CZ-O distance. (**f**) Benzene-CB-CA angle. (**g**) Overlap of _A_Y56. The dimer and S systems are shown in blue and red, respectively.

**Figure 13 ijms-22-04766-f013:**
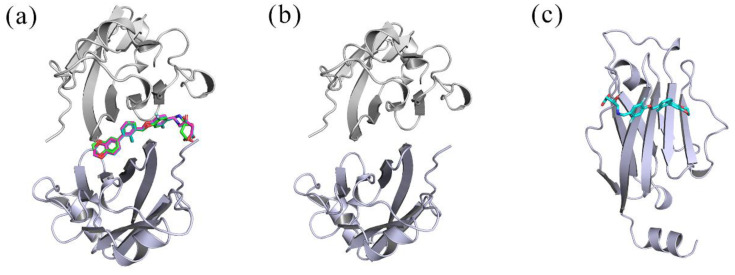
The initial structures of the systems used in MD simulation. (**a**) MOD, S, and R systems. (**b**) Dimer. (**c**) _A_PD-L1/(S)-BMS-200. The (MOD)-BMS-200, (S)-BMS-200 and (R)-BMS-200 components are shown in magenta, cyan and green, respectively.

**Table 1 ijms-22-04766-t001:** Binding free energies in the MOD, S, R, dimer and _A_PD-L1/(S)-BMS-200 systems (kcal/mol).

Contribution	MOD	S	R	Dimer	_A_PD-L1/ (S)-BMS-200
Δ*E*_vdw_ ^a^	−70.70 ± 0.40	−69.12 ± 0.59	−67.93 ± 0.99	−44.59 ± 9.85	−35.75 ± 2.02
Δ*E*_ele_ ^b^	−12.88 ± 1.00	−9.96 ± 0.78	−12.15 ± 0.56	−124.35 ± 23.36	−6.33 ± 0.99
Δ*E*_PB_ ^c^	47.06 ± 1.38	42.55 ± 1.46	45.54 ± 1.81	211.28 ± 17.07	25.67 ± 2.70
Δ*E*_SA_ ^d^	−5.92 ± 0.04	−5.88 ± 0.06	−5.95 ± 0.04	−6.23 ± 0.37	−3.78 ± 0.18
Δ*E*_polar,total_ ^e^	34.17 ± 2.38	32.59 ± 2.24	33.39 ± 2.37	86.94 ± 10.00	19.34 ± 0.92
Δ*E*_nonpolar_,_total_ ^f^	−76.62 ± 0.44	−75.00 ± 0.65	−73.88 ± 1.03	−50.82 ± 10.06	−39.53 ± 3.68
ΔG ^g^	−42.45 ± 0.35	−42.42 ± 0.21	−40.48 ± 0.21	36.11 ± 0.89	−20.17 ± 2.20

^a^ van der Waals interaction energy. ^b^ Electrostatic energy. ^c^ Polar solvent effect energy. ^d^ Nonpolar solvent effect energy. ^e^ Polar binding free energy. ^f^ Nonpolar binding free energy. ^g^ Binding free energy. The energies are the average values of the 300 conformations extracted from 120 to 150 ns.

**Table 2 ijms-22-04766-t002:** Contact numbers in the binding domains of the S and R systems.

Inhibitor	N-Terminal	Loop	Total Sheet	Total
(S)-BMS-200	33	18	457	507
(R)-BMS-200	22	24	446	492

**Table 3 ijms-22-04766-t003:** Hydrogen bond occupancies of the S system.

Donor	Donor H	Acceptor	Occupancy (%)
_A_Phe19@N	H	(S)-BMS-200@O5	28.57
_A_Phe19@N	H	(S)-BMS-200@O4	22.92
_A_Phe19@N	H	(S)-BMS-200@O3	0.33
_B_His69@NE2	HE2	(S)-BMS-200@O5	1.33
_B_His69@NE2	HE2	(S)-BMS-200@O4	1.33
_B_His69@NE2	HE2	(S)-BMS-200@O3	1.33

**Table 4 ijms-22-04766-t004:** Hydrogen bond occupancies of the R system.

Donor	Donor H	Acceptor	Occupancy (%)
(R)-BMS-200@O3	H3	_A_Asp122@OD2	42.52
_A_Lys124@NZ	HZ2	(R)-BMS-200@O4	4.98
_A_Lys124@NZ	HZ3	(R)-BMS-200@O5	2.99
_B_Lys75@NZ	HZ2	(R)-BMS-200@O4	1.66
_B_Gln66@NE2	2HE2	(S)-BMS-200@O5	0.66

## Data Availability

The data presented in this study are available within the article, figures, tables and Appendix A.

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
