# Peer review of "Molecular Mechanism of Small-Molecule Inhibitors in Blocking the PD-1/PD-L1 Pathway through PD-L1 Dimerization"

_ijms, 2021, doi:10.3390/ijms22094766_

Round 1
Reviewer 1 Report
In this manuscript, the authors used multiple molecular modelling methods together with the high-resolution crystal structure to present a detailed comprehensive insight into the inhibitory mechanism of BMS-200-related small-molecule inhibitors in the systems with the programmed cell death protein 1 (PD-1) and the programmed death-ligand 1 (PD-L1) dimer. The studies included the original (S)-BMS-200, its R-enantiomer (R)-BMS-200, and the (MOD)-BMS-200 without chirality by replacing the hydroxyl with carbonyl at the chiral center of (S)-BMS-200. The authors employed molecular modeling with Gauss View and optimized the minimized energy by Chem 3D software with MM2 force field. They used AutoDock Vina for
automatic placement of (R)-BMS-200 and (MOD)-BMS-200 in the binding pocket of the PD-L1 dimer to obtain the initial structures of the R and MOD systems. The authors then performed extensive molecular dynamics simulations as follows: General Amber Force Field, Amber ff99SB force field, TIP3P water model, GROMACS 2016.4 molecular dynamics package with heating in the NVT ensemble (1 ns) and then productive simulation in the NPT ensemble (150 ns, which is at the lower edge of duration needed to get meaningful results) with the Particle Mesh Ewald method. After the MD simulations, they calculated the binding free energies of these systems using the MM-PBSA approach in the GROMACS 2016.4 package and performed per-residue energy decomposition. The authors found that the formation of the PD-L1 dimer is greatly promoted by the BMS-200-related inhibitors, and the MOD system exhibits the same binding free energy as the S system, while the R system has a slightly higher value and the key residues are Ile54, Tyr56, Met115, Ala121 and Tyr123 ones on both the PD-L1 monomers of these systems. They also identified three main binding domains including the C, F and G strands of the PD-L1 dimer, found the dominant role of nonpolar interactions in stabilizing the S and R systems. The authors also found that compared to (S)-BMS-200, (R)-BMS-200 tends to form H bonds with charged residues. It followed from the authors' FEL and PPI analysis that (S)-BMS-200 could bind with the PD-L1 dimer stably and induce significant conformational changes of the key residues on the PD-L1 dimer, accelerating the compact interactions. The authors concluded that this study could be useful for structural modification and design of small-molecule inhibitors targeting the PD-1/PD-L1 pathway.
This manuscript presents a sound work theoretical modeling work that should be of interest to a wide community of researchers in biomolecular sciences. The manuscript can be published after the authors correct their numerous errors in manuscript grammar and add de-abbreviations of the numerous acronyms they used in the manuscript.
Reviewer 2 Report
In this manuscript, the authors characterize the inhibitory activity of (S)-BMS-200, (R)-BMS-200, (MOD)-BMS-200 with PD-L1 dimers complexes, in an effort to improve the understanding of the inhibitory mechanism of these inhibitors.
In brief, the authors conduct a 150ns simulation of various models of
PD-L1 with S)-BMS-200, (R)-BMS-200, (MOD)-BMS-200. They
present data to say that the BMS-200 structure significantly promotes dimerization. But the MM-GBSA results say otherwise. While these observations suggest the results, but they are far from definitive. It is important to note that the conclusions that one can make from a straightforward short molecular
dynamics simulation (especially since the authors do not use any enhanced sampling techniques to more fully explore the conformational landscape). Putting aside the issue of protein-ligands stability and the dimer formation, all of the observations arising from
these 150ns simulations are relatively obvious;
The grammar in the paper is particularly poor, which further makes understanding the importance of the results difficult. It is recommended that the manuscript be reviewed thoroughly for clarity and grammar, and that specific emphasis is placed on what is unique about the study and what is important about the results.
